# Direct healthcare costs of lip, oral cavity and oropharyngeal cancer in Brazil

**Vanessa Milani[1], Ana Laura de Sene Amâncio Zara[1], Everton Nunes da Silva [2‡], Larissa Barbosa Cardoso[3‡], Maria Paula Curado[4‡], Rejane Faria Ribeiro-Rotta [1] \***

**1** School of Dentistry, Universidade Federal de Goiás (UFG), Goiânia, Goiás, Brazil, **2** Universidade de Brasília (UnB), Brasília, Federal District, Brazil, **3** Faculty of Administration, Accounting and Economic Sciences, Universidade Federal de Goiás (UFG), Goiânia, Goiás, Brazil, **4** A. C. Camargo, São Paulo, São Paulo, Brazil

☉ These authors contributed equally to this work.
‡ These authors also contributed equally to this work.
\* rejanefrr@ufg.br

**Data Availability Statement:** Data are available from Figshare (DOI: 10.6084/m9.figshare. 13641449.v1).

## Abstract

The efficiency of public policies includes the measurement of the health resources used and their associated costs. There is a lack of studies evaluating the economic impact of oral cancer (OC). This study aims to estimate the healthcare costs of OC in Brazil from 2008 to 2016. This is a partial economic evaluation using the gross costing top-down method, considering the direct healthcare costs related to outpatients, inpatients, intensive care units, and the number of procedures, from the perspective of the public health sector. The data were extracted from the Outpatient and Inpatient Information System of the National Health System, by diagnosis according to the 10th Revision of the International Classification of Diseases, according to sites of interest: C00 to C06, C09 and C10. The values were adjusted for annual accumulated inflation and expressed in 2018 I\$ (1 I\$ = R\$2,044). Expenditure on OC healthcare in Brazil was I\$495.6 million, which was composed of 50.8% (I\$251.6 million) outpatient and 49.2% (I\$244.0 million) inpatient healthcare. About 177,317 admissions and 6,224,236 outpatient procedures were registered. Chemotherapy and radiotherapy comprised the largest number of procedures (88.8%) and costs (94.9%). Most of the costs were spent on people over 50 years old (72.9%) and on males (75.6%). Direct healthcare costs in Brazil for OC are substantial. Outpatient procedures were responsible for the highest total cost; however, inpatient procedures had a higher cost per procedure. Men over 50 years old consumed most of the cost and procedures for OC. The oropharynx and tongue were the sites with the highest expenditure. Further studies are needed to investigate the cost per individual, as well as direct non-medical and indirect costs of OC.

## Introduction

Oral cancer (OC) is a term used to classify a comprehensive group of neoplasms that can affect the lip, oral cavity, and oropharynx. OC is a current issue worldwide. Inequalities in the access to diagnostic and treatment services aggravate the prognosis and compromise the survival of

**Funding:** The author(s) received no specific funding for this work.

**Competing interests:** The authors have declared that no competing interests exist.

individuals affected by this disease, especially in lower middle- and upper middle-income countries, including most of the countries in South America, such as Argentina, Bolivia, Colombia, Peru and Brazil. In Brazil, around 15,190 new OC cases are estimated for 2021, making it the fifth most frequent cancer in men and the thirteenth in women, with incidence rates that increase with age, reaching a peak between 60 and 70 years old [1–7].

The world's demographic transition calls attention to the increase in noncommunicable diseases, such as OC, as it has a direct impact on the budget planning of health care [8]. The search for efficiency of public policies includes measurement of the health resources used and their associated costs [9]. Despite this, there is a large gap in the scientific literature in terms of studies evaluating the economic impact of OC [2].

Cost-of-illness studies on OC have gained increasing importance, since despite its low incidence this disease is often detected in advanced stages, when treatment is more complex and costlier. OC has been considered a disease with high mortality rates and high economic and social impact [10–12]. In the United Kingdom, the cost of treating an individual with OC can range from U$3,343 (I$6,133) in the early stages to U$24,890 (I$45,664) in advanced stages [13].

A cost-of-illness study on OC would be important to draw the attention of managers to the need for more assertive strategies, aimed at prevention, focusing on risk factors and detection of OC in the early stages, in order to reverse the current status: advanced disease as the most frequent diagnosis.

This study aims to estimate healthcare costs related to lip cancer (LC), oral cavity cancer (OCC) and oropharyngeal region cancer (OPC) in Brazil from 2008 to 2016.

## Materials and methods

### Study design

We conducted a prevalence-based cost-of-illness study from the perspective of the public healthcare system (Ministry of Health), with a top-down approach, from January 2008 to December 2016. We defined prevalence-based as all OC patients were included, regardless of the level of severity or onset of the disease. A top-down approach was used because our estimates consist of allocating total national health care expenditures by type of care (inpatient and outpatient) among the OC sites by using the ICD-10 codes for each site of OC (i.e. we identified the portions of the total health expenditure due to OC sites).

All codes related to OC according to the 10th Revision of the International Classification of Disease [ICD-10]) were included. Three anatomical regions were considered, as follows: **lip** (C00—malignant neoplasm of the lip), **oral cavity** (C02—malignant neoplasm of other and unspecified parts of the tongue, C03—malignant neoplasm of the gums, C04—malignant neoplasm of the floor of the mouth, C05—malignant neoplasm of the palate, and C06—malignant neoplasm of other and unspecified parts of the mouth) and the **oropharyngeal region** (C01—malignant neoplasm of the base of the tongue, C09—malignant neoplasm of the tonsil, and C10—malignant neoplasm of the oropharynx) [14].

### Brazilian health system

Brazil has the largest territorial extension of Latin America, made up of 27 Federation Units, subdivided into five regions (North, Northeast, South, Southeast and Midwest). The Brazilian population is covered by a Unified Health System popularly called SUS, which is the official Brazilian health system, in force since 1988. This system is based on the principles of universality, equity and integrality to provide full access to health at all levels of care, and considers health as a fundamental human right of the human being, which must be guaranteed by the

State. About 167 million people, 80% of the Brazilian population, depend exclusively on SUS for healthcare assistance [15].

## Data sources

Two important Brazilian health information systems are the Inpatient Information System (SIH-SUS) and the Outpatient Information System (SIA-SUS), tools used to record all hospitalisation and outpatient information, respectively, within the scope of SUS. Health facilities (hospitals, urgent care, clinics, pharmacies, etc.) included all procedures delivered to a patient during an outpatient or inpatient after his/her discharge, including health workers services. Each procedure is valued based on a fixed value defined by the Ministry of Health of Brazil. The sum of all procedures during as an outpatient or inpatient is sent to the Ministry of Health by means of reimbursement. On this basis, it represents the actual amount reimbursed to the health care providers for each outpatient procedure or inpatient stay. It is worth noting that each patient may have a different combination of procedures depending on disease severity.

Both systems use the ICD-10 and were conceived and implemented, initially, as a means of administrative financial control for the payment of services hired by SUS [16]. Despite this, these systems bring together demographic, geographic and diagnostic information and the consumption of healthcare resources for millions of users, at a low operating cost, which increases the possibilities for use in health evaluation, research, planning and management, besides providing information on the epidemiological profile of the population and the place of origin and care [16,17].

## Data analysis

The data were extracted and processed using Tab Win® software, version 4.15, an open access data tab, from the public health system. The expenses were classified by year (2008–2016), OC site (LC, OCC, OPC), outpatient and inpatient unit, sex (male or female), age group ($\leq$ 40 years, 41–50 years, 51–60 years, 61–70 years, and $>$ 70 years) and Brazilian region. Individual patient data were not available, and the expenses were aggregated and associated with the inpatient/outpatient admission documents.

For inpatient care expenses, the following direct healthcare costs were obtained: hospitalisation (e.g. daily rate, room fees, food, personal hygiene, bed support, hospital supplies, allied healthcare professional service costs, medications, and diagnostic and therapeutic auxiliary services), medical professional service costs, intensive care unit (ICU) costs (including the use of all equipment for the ICU, technical teams and 24h patient monitoring). We identified just one direct non-healthcare cost related to companion daily stay. As all expenses are aggregated within the inpatient system, the discrimination of each category was not possible.

For outpatient care expenses, resource use (quantity) and costs were considered for all direct healthcare costs such as ambulatory services and procedures, such as diagnostic imaging and laboratory tests (e.g. computed tomography, scintigraphy, magnetic resonance, biopsy, mammography, immunohistochemistry of malignant neoplasms, etc.), clinical procedures (e.g. chemotherapy, radiotherapy, cobalt therapy, radiotherapy imaging verification, radiotherapy and collimation planning, etc.), surgical procedures, orthoses and prostheses, complementary actions (e.g. food) and other procedures registered in the system. We also included non-direct healthcare costs related to transport for patients and/or companion.

Total expenditure was calculated in the Brazilian currency (Real [R$]) and converted into international dollars (I$) using the World Bank exchange rate for 2018 (1 I$ = R$2,044) [18], according to World Health Organization (WHO) recommendations [19]. Costs were updated to 2018 by applying the official rates of annual accumulated inflation (IPCA—Broad Consumer Price Index). Data were analysed using Microsoft® Excel® 2018.

The trend analysis of OC costs, by anatomical site, was performed using the Prais-Winsten method of linear regression, considering the trend analysis to model the seasonality that is present in the time series data of this study. The Prais-Winsten method is recommended in this type of analysis, especially to deal with the serial autocorrelation present in these cases. The serial autocorrelation can induce misinterpretation resulting from undue significance of minor variations. The smaller the number of points included in the series, the more sensitive this effect will be [20].

A critical p-value of 0.05 was adopted to determine a significant trend. Regression analysis was performed using STATA 14.0 software (StataCorp. 2015. Stata Statistical Software: Release 14. College Station, TX: StataCorp LP). The annual increment fee was calculated using Microsoft® Office Excel® 2018 software.

## Compliance with ethical standards

This study is exempt from evaluation by the Research Ethics Committee; informed consent was not needed due to all data being public, from secondary sources, open access and with no identification of subjects [21].

## Results

Expenditure on OC healthcare in Brazil, from 2008 to 2016, was I$495.6 million, which was composed of 50.8% outpatient and 49.2% inpatient expenditure. The oropharynx (21.9%), floor of the mouth (17.8%) and base of the tongue (17.2%), which is also part of the oropharyngeal anatomical region, were the sites with the highest expenditure (Table 1).

Over the investigated period, the annual incremental rate of expenditure was stationary for all sites (LC, OCC, base of the tongue, and tonsil) except for the oropharynx (C10). For the oropharynx, the increase rate was 5.9% per year (95% CI 1.4–10.6; beta = 0.0249; $p < 0.05$) (Table 1).

In the period 2008–2016, 177,317 hospital admissions were registered. I$118.0 million was spent on professional services (48.4%), I$92.5 million on hospital services (37.9%) and I$33.5 million on the ICU (13.7%) (Table 1).

The total number of outpatient procedures and services registered was 6,224,236, totalling I$251.7 million spent. Clinical procedures accounted for the largest numbers in quantity 93.7% (n = 5,831,242) and in expense 95.8% (I$241.1 million). Of the clinical procedures, radiotherapy and related procedures (cobalt therapy, radiotherapy and collimation planning, and radiotherapy imaging verification), together with chemotherapy also added up to the largest number of procedures (88.8%) and costs (94.9%) (Table 2).

The highest consumption of procedures, both outpatient and inpatient, and their associated costs were by male patients (75.6% of the cost). The total cost ratio/1,000 inhabitants was I$4,006 among men and I$1,241 among women. For both men (67.9%) and women (73.3%), spending was higher for those with OCC (Table 3).

Most of the costs (72.9%) were spent on individuals over 50 years old. The lowest cost ratio/1,000 inhabitants was among individuals under 40 years old (I$364.4/1,000 inhabitants) and the highest was in the age group 61 to 70 years old (I$11,317/1,000 inhabitants) (Table 3).

The lowest expenditure was in the North region of Brazil (2.1%) and the highest was in the Southeast region of the country (48.1%) (Table 3).

## Discussion

The main contribution of this study was to estimate the costs of OC in Brazil (I$495.6 million), which has not been reported before. This finding corresponds to an annual average cost of I$55.1 million (SD ± 5.0 million). Differences in the costing method and in the organisation of

**Table 1. Healthcare expenditure on inpatient and outpatient care for oral cancer in Brazil, 2008–2016.**

| Malignant neoplasm site | Inpatient | | | | | Outpatient | | Total costs (inpatient + outpatient)—I$ million (%) | Regression (beta) | Annual incremental ratio | | | Trend |
|---|---|---|---|---|---|---|---|---|---|---|---|---|---|
| | Admission (n) | Professional costs—I$ million (%) | Hospital service costs—I$ million (%) | Intensive care unit costs—I$ million (%) | Total costs—I$ million (%) | Procedures (n) | Total costs—I$ million (%) | | | Ratio (%) | 95% CI | | |
| C00 –Lip | 27,632 | 9.2 (50.8) | 7.9 (43.7) | 1.0 (5.5) | 18.1 (79.7) | 311,019 | 4.6 (20.3) | 22.7 (4.6) | −0.0053 | −1.20 | −5.54 | 3.34 | Stationary |
| LC | 27,632 | 9.2 (50.8) | 7.9 (43.7) | 1.0 (5.5) | 18.1 (79.7) | 311,019 | 4.6 (20.3) | 22.7 (4.6) | - | - | - | - | - |
| C02—Other and unspecified parts of the tongue | 32,933 | 17.8 (47.3) | 14.9 (39.6) | 4.9 (13.1) | 37.6 (46.4) | 1,121,205 | 43.5 (53.6) | 81.1 (16.4) | 0.0021 | 0.48 | −6.10 | 7.52 | Stationary |
| C03 –Gum | 2,148 | 1.6 (55.2) | 1.1 (37.9) | 0.2 (6.9) | 2.9 (39.7) | 113,519 | 4.6 (61.3) | 7.5 (1.5) | 0.0087 | 2.01 | −4.87 | 9.39 | Stationary |
| C04—Floor of the mouth | 19,256 | 34.2 (52.6) | 26.9 (41.3) | 4.0 (6.1) | 65.1 (73.6) | 596,597 | 23.4 (26.4) | 88.5 (17.9) | 0.0092 | 2.14 | −6.68 | 11.80 | Stationary |
| C05 –Palate | 7,670 | 3.0 (51.7) | 2.4 (41.4) | 0.4 (6.9) | 5.8 (21.0) | 519,107 | 21.8 (79.0) | 27.6 (5.6) | 0.0207 | 4.89 | −2.33 | 12.64 | Stationary |
| C06—Other and unspecified parts of the mouth | 22,825 | 13.0 (46.9) | 9.9 (35.9) | 4.8 (17.2) | 27.7 (52.9) | 649,690 | 24.7 (47.1) | 52.4 (10.6) | 0.0037 | 0.86 | −5.50 | 7.66 | Stationary |
| OCC | 84,832 | 69.6 (50.0) | 55.2 (39.7) | 14.3 (10.3) | 139.1 (54.1) | 3,000,118 | 118.0 (45.9) | 257.1 (51.9) | - | - | - | - | - |
| C01—Base of the tongue | 25,690 | 20.0 (39.9) | 15.4 (30.8) | 14.7 (29.3) | 50.1 (58.8) | 662,749 | 35.1 (41.2) | 85.2 (17.2) | 0.0237 | 5.61 | −1.90 | 13.69 | Stationary |
| C09 –Tonsil | 4,698 | 1.8 (51.4) | 1.4 (40.0) | 0.3 (8.6) | 3.5 (15.8) | 399,711 | 18.6 (84.2) | 22.1 (4.5) | 0.0194 | 4.56 | −2.25 | 11.84 | Stationary |
| C10 –Oropharynx | 34,465 | 17.3 (52.1) | 12.6 (38.0) | 3.3 (9.9) | 33.2 (30.6) | 1,850,639 | 75.3 (69.4) | 108.5 (21.9) | 0.0249 | 5.90 | 1.38 | 10.62 | Growing |
| OPC | 64,853 | 39.1 (45.0) | 29.4 (33.9) | 18.3 (21.1) | 86.8 (40.2) | 2,913,099 | 129.0 (59.8) | 215.8 (43.5) | - | - | - | - | - |
| Total | 177,317 | 117.9 (48.3) | 92.5 (37.9) | 33.6 (13.8) | 244.0 (49.2) | 6,224,236 | 251.6 (50.8) | 495.6 (100.0) | - | - | - | - | - |

health systems are two of the main reasons that makes economic evaluation results comparisons a challenge. Evidence has shown an increase in the number of health economic evaluations (HEEs) of cancer in Brazil [4]. However, although oral cancer is one of the five most common neoplasms among males, only 1.8% of the HEEs found were related to the oral cavity or pharyngeal cancer, from 1998 to 2013 [4] and none of them was a cost-of-illness study. We did not find any specific study on the cost of oral cancer. The only cost-of-illness study found in the head and neck region, with epidemiological (incidence rate) and methodological (same methodology of aggregate cost from the information systems of the Ministry of Health of Brazil, using ICD-10 code, 2008–2015) similarities to ours, it was related to thyroid cancer [22], for which the therapeutic approach is very different, making the costs incomparable.

For OC, the number of procedures was considered high, with an annual average of 691.6 thousand. Even though outpatient procedures presented a higher total value, the cost per

**Table 2. Most common procedures and services and their associated costs adopted during outpatient oral cancer care in Brazil, 2008–2016.**

| Nature of procedure | Total procedures | | Most used procedures | Quantity | | Costs | | Cost per procedure |
|---|---|---|---|---|---|---|---|---|
| | n | I$ million* | | n | %** | I$ million* | %*** | I$ million |
| Diagnostic | 149,400 | 7.7 | Computed tomography | 143,471 | 2.3 | 4.9 | 2.0 | 20.583,0 |
| | | | Scintigraphy | 3,557 | 0.1 | 0.5 | 0.2 | 510,3 |
| | | | Magnetic resonance | 2,547 | 0.0 | 0.5 | 0.2 | 365,4 |
| | | | Biopsy | 20,847 | 0.3 | 0.4 | 0.1 | 2.990,8 |
| | | | Mammography | 9,272 | 0.1 | 0.4 | 0.1 | 1.330,2 |
| | | | Immunohistochemistry of malignant neoplasms | 4,487 | 0.1 | 0.3 | 0.1 | 643,7 |
| | | | *Total diagnostic* | *184,181* | *3.0* | *7.0* | *2.8* | *740,7* |
| Clinical | 5,831,242 | 241.1 | Chemotherapy | 177,658 | 2.9 | 128.7 | 51.1 | 743,9 |
| | | | Radiotherapy | 3,742,209 | 60.1 | 73.8 | 29.3 | 15.668,7 |
| | | | Cobalt therapy | 1,279,969 | 20.6 | 23.2 | 9.2 | 5.359,2 |
| | | | Radiotherapy and collimation planning | 209,936 | 3.4 | 9.0 | 3.6 | 879,0 |
| | | | Radiotherapy imaging verification | 115,309 | 1.9 | 4.0 | 1.6 | 482,8 |
| | | | *Total clinical* | *5,525,081* | *88.8* | *238.8* | *94.9* | *22220,5* |
| Surgery | 96,989 | 1.5 | Excision | 58,331 | 0.9 | 1.1 | 0.4 | 38.946,1 |
| | | | Abscess drainage | 29,650 | 0.5 | 0.3 | 0.1 | 19.796,6 |
| | | | Grade II dressing | 5,126 | 0.1 | 0.1 | 0.0 | 3.422,5 |
| | | | *Total surgery* | *93,107* | *1.5* | *1.5* | *0.6* | *374,5* |
| Orthoses and prostheses | 4,800 | 0.1 | Prostheses | 3,264 | 0.1 | 0.1 | 0.0 | 27.420,2 |
| | | | Colostomy/urostomy plate and/or pouch | 1,420 | 0.0 | 0.0 | 0.0 | 11.929,1 |
| | | | *Total orthoses and prostheses* | *4,684* | *0.1* | *0.1* | *0.0* | *18,8* |
| Complementary actions | 141,805 | 1.2 | Allowance for patient/companion food | 22,747 | 0.4 | 0.3 | 0.1 | 18.546,5 |
| | | | Patient/companion transport/displacement | 119,058 | 1.9 | 0.9 | 0.4 | 97.072,6 |
| | | | *Total complementary actions* | *141,805* | *2.3* | *1.2* | *0.5* | *570,3* |
| **Total** | **6,224,236** | **251.7** | **Total (most used)** | **5,948,858** | **95.6** | **248.6** | **98.8** | **23.924,9** |

*Costs are presented in I$ million dollars (currency: 1 I$ = R$2.044) (The World Bank Group, 2018) [18].

**Quantity (%) percentage relative to the total number of procedures reported during outpatient care in the period.

***Cost (%) percentage relative to the total outpatient costs in the period.

Italic markings represent the partial sum for each category, and bold markings represent the total sum.

procedure was 34 times higher for inpatient (I$1,375) than outpatient procedures (I$40). In Brazil, studies have shown that most patients are diagnosed with oral cancer at an advanced stages (III and IV), and in these cases the treatment includes more procedures and demand more resources [23,24]. Treatment costs for cancer may vary according to the stage of disease. Souza et al. (2009) showed that the cost of illness for skin cancer in Brazil ranges from I$187.3 (R$382.8) in stage 0 to I$15,667 (R$32,024) in stage IV [25]. In the United Kingdom, the average of outpatient and inpatient costs per patient, after one year of treatment, were US$3,343 (I$6,133) for precancerous lesions and US$24,890 (I$45,664) for cases in stage IV [13].

Cancer staging may also influence the length of hospital stay, with those diagnosed with potentially malignant disorders remaining in hospital for 1.9 days on average, while individuals with a late diagnosis (stage IV) remain in hospital for 29.9 days [13]. Although our results do not explicitly state this cost by stage, based on this evidence it is possible assume that, the more advanced the disease, the more inpatient procedures will be necessary, which substantially increases the cost of the illness. Part of these efforts could be better allocated to other diseases if the incidence and severity of OC were reduced.

**Table 3. Healthcare procedures, admissions and their associated expenditure (I$ million) for inpatient and outpatient care for oral cancer in Brazil, by site group, sex and age group in Brazil, 2008–2016.**

| Variable | | Outpatient | | | Inpatient | | | Total | |
|---|---|---|---|---|---|---|---|---|---|
| | | Procedures (n) | Cost (I$ million)* | % Cost | Admissions (n) | Cost (I$ million)* | % Cost | Cost (I$ million)* | % Cost |
| **Sex**** | | | | | | | | | |
| | **Male** | | | | | | | | |
| | LC | 153,883 | 3.0 | 1.5 | 15,426 | 10.5 | 6.2 | 13.6 | 3.6 |
| | OCC | 2,228,526 | 91.9 | 45.3 | 60,352 | 101.0 | 58.8 | 192.9 | 51.6 |
| | OPC | 1,852,106 | 107.7 | 53.2 | 50,253 | 60.0 | 35.0 | 167.7 | 44.8 |
| | Total | 4,787,409 | 202.6 | 80.8 | 126,031 | 171.5 | 70.0 | 374.2 | 75.5 |
| | **Female** | | | | | | | | |
| | LC | 157,136 | 1.6 | 3.3 | 12,206 | 7.4 | 10.0 | 9.0 | 7.4 |
| | OCC | 771,592 | 26.7 | 55.4 | 24,480 | 38.2 | 52.0 | 64.9 | 53.3 |
| | OPC | 480,936 | 19.9 | 41.3 | 14,600 | 27.9 | 38.0 | 47.8 | 39.3 |
| | Total | 1,409,664 | 48.2 | 19.2 | 51,286 | 73.5 | 30.0 | 121.7 | 24.5 |
| **Age group (years)**** | | | | | | | | | |
| | ≤ 40 | 377,067 | 14.6 | 5.9 | 21,744 | 34.7 | 14.3 | 49.2 | 10.1 |
| | 41–50 | 1,114,455 | 46.7 | 19.0 | 26,278 | 36.6 | 15.1 | 83.3 | 17.1 |
| | 51–60 | 2,161,502 | 90.6 | 36.8 | 71,867 | 67.7 | 27.9 | 158.3 | 32.4 |
| | 61–70 | 1,529,599 | 60.8 | 24.7 | 34,174 | 56.3 | 23.2 | 117.0 | 24.0 |
| | > 70 | 1,014,450 | 33.4 | 13.6 | 23,254 | 47.1 | 19.4 | 80.5 | 16.5 |
| **Brazilian region**** | | | | | | | | | |
| | Northern | 183,853 | 6.5 | 2.6 | 3,354 | 3.2 | 1.5 | 9.7 | 2.1 |
| | Northeast | 1,330,370 | 49.1 | 19.6 | 48,928 | 61.4 | 29.2 | 110.5 | 24.0 |
| | Southeast | 3,281,996 | 133.1 | 53.1 | 76,705 | 88.5 | 42.1 | 221.6 | 48.0 |
| | Southern | 1,101,947 | 47.9 | 19.1 | 36,478 | 44.3 | 21.1 | 92.2 | 20.0 |
| | Midwest | 298,907 | 14.2 | 5.7 | 11,852 | 13.0 | 6.2 | 27.2 | 5.9 |

LC: Lip cancer; OCC: Oral cavity cancer; OPC: Oropharyngeal cancer.

*Costs are presented in I$ million dollars (currency: 1 I$ = R$2.044) (The World Bank Group, 2018) [18].

** Missing values: 27,163 outpatient procedures.

Some anatomical sites of OCC, such as the tongue and floor of the mouth, are presented in the literature as having the highest incidence of OC [26–28]. This could explain this study's findings that this group was shown as the costliest for males and females and is also the one with highest demand for hospitalisation hours, attendants, and ICU. Also, this highest incidence regions are anatomically more complex and, considering that most cases of oral cancer that reach treatment are in advanced stages [29], the high cost of cancers in these regions may be related to a greater demand for more complex multimodal and surgical treatments, integrating multi- professional teams (vascular microsurgery, plastic surgery) and requiring professionals specialised in long-term rehabilitation [30].

The high OPC costs may be related to the rising incidence trend in OPC that has been observed in the recent decades, worldwide. The association of the burden of the disease in young patients and with human papillomavirus (HPV) infection [31] has required investments in new diagnostic and treatment approaches, which may also have contributed to the high cost of OPC. The treatment has become longer, and new radiotherapy techniques have been implemented to reduce the side effects of the treatment and the multidisciplinary team needed to learn more about the disease in the oropharynx. Individuals with HPV-positive OPC have responded better to treatment than others. These individuals have shown an average survival

of two years (87.5%-95.0%) higher than negative-HPV individuals (62.0%-67.2%) [32]. Different from others head and neck tumours, OPC associated with HPV tends to occur in the young population group (30 to 50 years old), in non-smokers and non-drinking individuals, and in people who are employed and have young families; consequently, this cancer has a considerable social and economic impact [33,34].The inclusion of the diagnosis of HPV [35] as a routine and the need for training of pathologists for this diagnosis, also need to be highlighted as possible contributors to the high cost of OPC.

Tobacco smoking and alcohol drinking are responsible for a large proportion of oral and pharyngeal cancers and account for a higher proportion of head and neck cancers among men than among women. Even though the data collection was not based on individuals, the results reproduce the epidemiological aspects of OC [3,9,10,26,36–39], since the number of procedures and cost of illness were three times higher for men over 50 years of age than for women.

Regarding Brazilian regions, three of them (South, Southeast and Northeast) consumed 92% of the total costs for OC. This distribution coincides with the highest proportion of smokers, which is also concentrated in these three Brazilian regions [40]. The Southeast region received approximately 50% of the total resources. Although the costs of OC by region are proportional to the population distribution in them, these results might be related to better access to technologies, greater service offerings, greater population access to the health system and a more effective systematisation of SUS notifications [41].

In terms of the limitations of this study, we highlight the below aspects, which could be considered in the planning for further studies: the SIA-SUS and SIH-SUS data do not allow for identifying the OC costs by individual; the number of procedures and actual costs tend to be underestimated, since the data used was only from public health services and did not include the private system data. Another important aspect to be considered for further studies is presenting costs in international dollars, as it is recommended by WHO [19], since it would be a trying to equalise the purchasing power of different currencies, by eliminating the differences in price levels between countries.

According to the WHO (2018) [42], a strategy for reducing OC incidence as a noncommunicable disease must include cost-effective interventions and be affordable, feasible and scalable in all settings. Examples of these would be to enact and enforce effective anti-smoking and alcohol-related harm public policies; vaccinate 9 to 13-year-old girls against human papillomavirus and prevent it by screening [42]. In addition, OC organized screening programs in lower middle- and upper middle-income countries can be ineffective in achieving the desired impact or suffer from poor quality of testing and follow up services. An alternative would be generating population awareness about the early detection of OC and training the frontline healthcare professional regarding the early symptoms, as an early detection approach [5,43].

## Conclusions

Direct healthcare costs for OC represented an average of I$55.1 million annually. Outpatient procedures were responsible for the highest total cost; however, inpatient procedures had a higher cost per procedure ratio. Men over 50 years old consumed most of the direct healthcare costs and procedures for OC, which is consistent with epidemiological data worldwide. The oropharynx and tongue were the anatomical sites with the highest costs, with the oropharynx presenting an annual growth trend. Further studies are needed to investigate cost per individual, direct non-medical costs and indirect costs for OC.

## Acknowledgments

We thank Dr Erika Carvalho Aquino for performing trend analysis of the linear regression.

## Author Contributions

**Conceptualization:** Vanessa Milani, Ana Laura de Sene Amâncio Zara, Rejane Faria Ribeiro-Rotta.

**Data curation:** Vanessa Milani, Ana Laura de Sene Amâncio Zara, Rejane Faria Ribeiro-Rotta.

**Formal analysis:** Vanessa Milani, Ana Laura de Sene Amâncio Zara, Rejane Faria Ribeiro-Rotta.

**Funding acquisition:** Larissa Barbosa Cardoso, Rejane Faria Ribeiro-Rotta.

**Investigation:** Vanessa Milani, Ana Laura de Sene Amâncio Zara, Rejane Faria Ribeiro-Rotta.

**Methodology:** Vanessa Milani, Ana Laura de Sene Amâncio Zara, Everton Nunes da Silva, Rejane Faria Ribeiro-Rotta.

**Project administration:** Vanessa Milani, Ana Laura de Sene Amâncio Zara, Rejane Faria Ribeiro-Rotta.

**Resources:** Rejane Faria Ribeiro-Rotta.

**Supervision:** Everton Nunes da Silva, Larissa Barbosa Cardoso, Maria Paula Curado.

**Visualization:** Vanessa Milani, Ana Laura de Sene Amâncio Zara, Rejane Faria Ribeiro-Rotta.

**Writing – original draft:** Vanessa Milani, Ana Laura de Sene Amâncio Zara, Rejane Faria Ribeiro-Rotta.

**Writing – review & editing:** Vanessa Milani, Ana Laura de Sene Amâncio Zara, Everton Nunes da Silva, Larissa Barbosa Cardoso, Maria Paula Curado, Rejane Faria Ribeiro-Rotta.

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
