## [Decision Letter · Decision Letter 0]

17 Sep 2020

PONE-D-20-19121

Direct healthcare costs of lip, oral cavity and oropharyngeal cancer in Brazil

PLOS ONE

Dear Dr. ROTTA,

Thank you for submitting your manuscript to PLOS ONE. After careful consideration, we feel that it has merit but does not fully meet PLOS ONE’s publication criteria as it currently stands. Therefore, we invite you to submit a revised version of the manuscript that addresses the points raised during the review process.

Below are the reviewers’ comments that I would like for you to address. The main concern from both reviewers relates to the methodology, which is not clear whether it is appropriate for your study. You must justify and explain better the implementation of your methods.

We look forward to receiving your revised manuscript.

Kind regards,

Gabriel A. Picone

Academic Editor

PLOS ONE

Journal Requirements:

2. In your Methods section, please provide additional details regarding the dates upon which these data sources were accessed.

Reviewers' comments:

Reviewer's Responses to Questions

**Comments to the Author**

1. Is the manuscript technically sound, and do the data support the conclusions?

Reviewer #1: Partly

Reviewer #2: Partly

2. Has the statistical analysis been performed appropriately and rigorously? 

Reviewer #1: No

Reviewer #2: No

3. Have the authors made all data underlying the findings in their manuscript fully available?

Reviewer #1: Yes

Reviewer #2: Yes

4. Is the manuscript presented in an intelligible fashion and written in standard English?

Reviewer #1: No

Reviewer #2: Yes

5. Review Comments to the Author

Reviewer #1: The study presents estimates of the direct medical expenditure incurred on oral cancer by the Brazilian health system. The study is really useful in defining the magnitude of the disease in terms of economic burden. Further, the estimates from the present study can also be used for assessing the efficiency of the present health system and strategies for reducing the health expenditure. Although, the study achieves its main objective of estimating the direct cost on OC, but I have a few concerns in the methodology followed:

1. The present analysis is actually a type of prevalence based cost of illness study. But, I am not confident (as per details mentioned in the methods section) that it uses the principles of top-down costing methods. The study simply measures the actual quantity of services delivered as the part of management of OC in Brazil (for the reference time period) and then multiply it with the cost of delivering these services. As the top down approach uses some type of aggregate data (morbidity, treatment utilization or total cost on health care delivery) which is then allocated to specific service or diseases. The authors should justify the use of top-down approach in the present study as the details of it in either data collection or data analysis is not clearly mentioned in the methods section.

2. The authors report that the data from Inpatient Information System (SIH-SUS) and the Outpatient Information System (SIA-SUS) was used to record hospitalization and outpatient information. It should be clearly mentioned that what type of information or data was assessed from the above mentioned resources. Whether only the utilization data on the number of hospitalizations, ICU admissions and outpatient procedures corresponding to the specific ICD-10 codes was extracted or the information on the specific resource use for each of the inpatient and outpatient procedure was also available from the data sources (as mentioned on the lines 114 -119 and 121-122). If the segregated information on resource use was not available (lines 119-120), how the inpatient costs were segregated into professional costs and hospital service costs/ICU costs (table 1).

3. Operational definition of professional cost (with regards to inpatient care) is missing (what cost components were included in this). ICU cost also seems to include some component of professional cost (cost of technical teams). Authors should take into consideration that is there is no duplication of costs.

4. The authors mentioned that expenses based on prices represented by the Brazilian Ministry of Health’s reimbursements to all health providers were used for estimating the cost. The manuscript lacks details on what type of information on prices was used from the reimbursement system.

a. Whether, an estimate of average unit cost on outpatient procedures or inpatient stay/ICU stay was extracted from the reimbursement system and simply multiplied with the number of procedures to estimate the total direct cost.

b. OR it is the actual amount reimbursed to the health care provider for each of the outpatient procedure or inpatient stay.

c. OR prices of various resources consumed in the delivery of outpatient and inpatient care were extracted from the reimbursement system and then these were used to calculate the cost the treatment given.

5. Transport cost and cost of companion is usually included in the category of direct non-health care cost. Author can think of adjusting this issue.

6. Is there any other reason for the higher cost in oropharynx/floor of mouth/base of tongue besides the higher incidence of cancer for these specific sites? Or there was more utilization of outpatient procedures and inpatient services for the management of these specific cancer sites as compared to other sites. Similarly, what is the possible reason of ‘growing’ expenditure in the case of oropharynx as compared to stationary rates for other sites? It is important and should be mentioned in the discussion section.

7. Authors have mentioned total number of procedures and total cost of each procedure in table 2. It will be also useful to add cost per procedure in a separate column in the table.

8. It should be mentioned in table 1 that the cost are presented in I$ million dollars.

9. The authors mention that costs for skin cancer in Brazil can vary highly according to the stage of disease. If possible, the authors should provide the segregation of total direct cost on OC as per stage of diagnosis of the disease. This will further strengthen the argument.

10. The line 249-250 i.e., ‘presentation of the SIA-SUS and SIH-SUS values by number of procedures is an impairment to identifying the individual costs of OC”. Please try to rephrase it for better understanding.

11. Overall, the discussion section needs improvement in terms of clarity of the language for better understanding of the readers.

Reviewer #2: The authors presented a paper about direct healthcare costs of lip, oral cavity and oropharyngeal cancer in Brazil. There is a large gap in the scientific literature on studies evaluating the economic impact of cancer treatment, especially in lower-middle- and upper-middle income countries. It is very important to disseminate information about this issues in regions where this information are rare. Particularly for OC, where the late detection enlarge exponentially costs and lives it is very important evaluate its economic impact to built efficient public policies.

I would like to contribute with some questions on intention to clarify some aspects of the paper

Major issues:

Methodology

1. Page 6, lines 133-139. Although the authors has explained the source of databases and how they extract the information, it is unclear the choice of this methodology and a theoretical support. When the authors use to calculate TCR the total cost in numerator, they are introducing an important bias in the results because they are treating procedures as individuals, wich produce an overestimation of the TCR. We know that the we have linked to an only person a set of procedures to the same treatment.

2.Page 7, lines 140-141. The authors should explain why they choose the Prais-Winsten method to perform linear regression and include more details about the model

Results

1. Page 9, lines 157-160 and Table 1. The parameters of the linear regression on table 1 is insufficient to conclude about the model adjustment and about the significance of the β parameters.

Discussion

1. Page 15, lines 194-202. It is difficult to compare both studies at least to reasons: the thyroid study use a methodology that really estimates a cost by patient. The other reason is despite of both topographies had similar incidence rates, the terapeutic approach is very different which made the costs incomparable.

2. Page 15, lines 203-206. The authors should take in mind the OC epidemiology. The incidence in males is about 3,5 times higher than in woman and this issue will be reproduced in the healthcare costs

3. Page 17, lines 236-247. The authors should take in mind that regional distribution follow the population distribution and it is not easy to link with the inequalities in public spending between the different regions

Minor issues:

Introduction:

1. Page 3, lines 50-51: Update the information about Brazilian cancer estimates to 2020-2022, like the authors already done in another part of the manuscript.

6. PLOS authors have the option to publish the peer review history of their article (what does this mean?). If published, this will include your full peer review and any attached files.

Reviewer #1: No

Reviewer #2: No

---

## [Author Response · Author response to Decision Letter 0]

28 Nov 2020

We carefully revised all the points raised during the manuscript review process, accepted and / or justify the reviewer' suggestions. The answers are described in detail in the rebuttal letter, and bellow (Figures inserted in the rebuttal letter could not be copied to here).

Reviewer 1, comment 1: “The study presents estimates of the direct medical expenditure incurred on oral cancer by the Brazilian health system. The study is really useful in defining the magnitude of the disease in terms of economic burden. Further, the estimates from the present study can also be used for assessing the efficiency of the present health system and strategies for reducing the health expenditure. Although, the study achieves its main objective of estimating the direct cost on OC, but I have a few concerns in the methodology followed:”

Our response: We really appreciate Reviewer’s 1 comments and time spent reviewing our manuscript.

Reviewer 1, comment 2:“1. The present analysis is actually a type of prevalence based cost of illness study. But, I am not confident (as per details mentioned in the methods section) that it uses the principles of top-down costing methods. The study simply measures the actual quantity of services delivered as the part of management of OC in Brazil (for the reference time period) and then multiply it with the cost of delivering these services. As the top down approach uses some type of aggregate data (morbidity, treatment utilization or total cost on health care delivery) which is then allocated to specific service or diseases. The authors should justify the use of top-down approach in the present study as the details of it in either data collection or data analysis is not clearly mentioned in the methods section.”

Our response: Thank you for raising this question and for the opportunity to clarify it. The bottom-up approach relies on individual data, obtained by multiplying the unit costs by quantities [Tarricone 2006]. This information is not publicly available in the information systems in Brazil (namely Inpatient and Outpatient Information Systems – SIH-SUS and SIA-SUS, respectively). In these information systems, the unit of analysis is the procedure instead of the individual. Moreover, patients can use different combinations of services (procedures) on the same inpatient or outpatient bill. Based on that, we cannot claim that our estimates are based on bottom-up approach. On the other hand, we used aggregate data to estimate the oral cancer (OC) costs. Our estimates consist of allocating total national health care expenditures by type of care (inpatient and outpatient) among the oral cancer sites by using the ICD-10 codes for each site of OC. In other terms, we identified the portions of the total health expenditure due to oral cancer sites. Moreover, we have not calculated the costs; we just extracted the aggregate costs from the information systems. We reworded the text in the manuscript to clarify it:

‘We defined prevalence-based as all OC patients were included, regardless of the level of severity or onset of the disease. A top-down approach was used because our estimates consist of allocating total national health care expenditures by type of care (inpatient and outpatient) among the OC sites by using the ICD-10 codes for each site of OC (i.e. we identified the portions of the total health expenditure due to OC sites).’ (Methods, Study design, page 4, line 76-80). 

Reference:

Tarricone R. Cost-of-illness analysis. What room in health economics? Health Policy. 2006 Jun;77(1):51-63. doi: 10.1016/j.healthpol.2005.07.016. Epub 2005 Sep 1. PMID: 16139925.

Reviewer 1, comment 3: “The authors report that the data from Inpatient Information System (SIH-SUS) and the Outpatient Information System (SIA-SUS) was used to record hospitalization and outpatient information. It should be clearly mentioned that what type of information or data was assessed from the above mentioned resources. Whether only the utilization data on the number of hospitalizations, ICU admissions and outpatient procedures corresponding to the specific ICD-10 codes was extracted or the information on the specific resource use for each of the inpatient and outpatient procedure was also available from the data sources (as mentioned on the lines 114 -119 and 121-122). If the segregated information on resource use was not available (lines 119-120), how the inpatient costs were segregated into professional costs and hospital service costs/ICU costs (table 1)”.

Our response: The issue raised by the reviewer is closely related to the previous comment. We apologise for not providing a complete picture of the Brazilian health information systems. We hope to clarify it at this stage of the review, as follows: 

‘Health facilities (hospitals, urgent care, clinics, pharmacies, etc.) included all procedures delivered to a patient during an outpatient or inpatient after his/her discharge, including health workers services. Each procedure is valued based on a fixed value defined by the Ministry of Health of Brazil. The sum of all procedures as an outpatient or inpatient is sent to the Ministry of Health by means of reimbursement. On this basis, it represents the actual amount reimbursed to the health care providers for each outpatient procedure or inpatient stay. It is worth noting that each patient may have a different combination of procedures depending on disease severity.’ (Methods, Data sources, page 5, line 104-112)

Reviewer 1, comment 4: “Operational definition of professional cost (with regards to inpatient care) is missing (what cost components were included in this). ICU cost also seems to include some component of professional cost (cost of technical teams). Authors should take into consideration that is there is no duplication of costs.

Our response: Thank you for your comment. As we stated by Reviewer 1, comment 2, we have not calculated the costs related to OC (prices multiplied by quantities). We collected the aggregate cost from the information systems of the Ministry of Health of Brazil, using ICD-10 codes for OC. For each procedure, the Ministry of Health defined a reimbursement fee, which has a fixed value. The Ministry of Health is the one that multiplies prices by quantities. On this basis, there is no duplication of cost. Regarding the definition of professional costs, the Ministry of Health of Brazil also defined a reimbursement fee for this purpose, which depends on the complexity of the procedure and the qualification of the professional team (Methods, Data sources, page 5, line 104-112).

Reviewer 1, comment 5: “The authors mentioned that expenses based on prices represented by the Brazilian Ministry of Health’s reimbursements to all health providers were used for estimating the cost. The manuscript lacks details on what type of information on prices was used from the reimbursement system. a. Whether, an estimate of average unit cost on outpatient procedures or inpatient stay/ICU stay was extracted from the reimbursement system and simply multiplied with the number of procedures to estimate the total direct cost. b. OR it is the actual amount reimbursed to the health care provider for each of the outpatient procedure or inpatient stay. c. OR prices of various resources consumed in the delivery of outpatient and inpatient care were extracted from the reimbursement system and then these were used to calculate the cost the treatment given.

Our response: Thank you again for the opportunity to clarify this issue. Option ‘b’ is correct as it is the actual amount reimbursed to the health care provider for each outpatient procedure or inpatient stay. This information was also included in the manuscript, in the Methods, Data sources, page 5, line 104-112, as was mentioned in response to comment 3 above. 

Reviewer 1, comment 6:“Transport cost and cost of companion is usually included in the category of direct non-health care cost. Author can think of adjusting this issue.

Our response: Thank you for suggesting this change, with which we totally agree. We have corrected it in the manuscript. (Methods, Data analysis, page 6, line 132; page 7, lines 142-143)

Reviewer 1, comment 7:“Is there any other reason for the higher cost in oropharynx/floor of mouth/base of tongue besides the higher incidence of cancer for these specific sites? Or there was more utilization of outpatient procedures and inpatient services for the management of these specific cancer sites as compared to other

sites. Similarly, what is the possible reason of ‘growing’ expenditure in the case of oropharynx as compared to stationary rates for other sites? It is important and should be mentioned in the discussion section.”

Our response: Thank you for your suggestion. We restructured the discussion session to address the suggested aspects. A paragraph was added in the discussion section, as follows:

‘Considering that most cases of oral cancer that reach treatment are in advanced stages [29] and that the floor of the mouth and the posterior boundary of the oral cavity are anatomically more complex, the high cost of cancers in these regions may be related to a greater demand for more complex multimodal and surgical treatments, integrating multi-professional teams (vascular microsurgery, plastic surgery) and requiring professionals specialised in long-term rehabilitation [30].’ (Discussion, page 18-19, lines 279-284)

Another paragraph was reformulated in the discussion section, as follows:

‘The high OPC costs may be related to the rising incidence trend in OPC that has been observed in recent decades worldwide. The association of the burden of the disease in young patients and with human papillomavirus (HPV) infection [31] has required investments in new diagnostic and treatment approaches, which may also have contributed to the high cost of OPC. The treatment has become longer, and new radiotherapy techniques have been implemented to reduce the side effects of the treatment and the multidisciplinary team needed to learn more about the disease in the oropharynx. Individuals with HPV-positive OPC have responded better to treatment than others. These individuals have shown an average survival of two years (87.5%-95.0%) higher than negative-HPV individuals (62.0%-67.2%) [32]. Different from other head and neck tumours, OPC associated with HPV tends to occur in the young population (30 to 50 years old), in non-smokers and non-drinking individuals, and in people who are employed and have young families; consequently, this cancer has a considerable social and economic impact [33, 34].The inclusion of the diagnosis of HPV [35] as routine and the need for training pathologists for this diagnosis also need to be highlighted as possible contributors to the high cost of OPC.’ (Discussion, pages 19-20, lines 296-312)

Reviewer 1, comment 8:“Authors have mentioned total number of procedures and total cost of each procedure in table 2. It will be also useful to add cost per procedure in a separate column in the table.”

Our response: Thank you for your suggestion. We have included this column in Table 2. (Results, page 11-12, Table 2)

Reviewer 1, comment 9:“It should be mentioned in table 1 that the cost are presented in I$ million dollars.”

Our response: Thank you for identifying this missing information. We have included ‘million’ in Table 1. Just to clarify, we have used international dollars, which take into consideration the Purchase Parity Power. (Results, page 9, Table 1).

Reviewer 1, comment 10:“The authors mention that costs for skin cancer in Brazil can vary highly according to the stage of disease. If possible, the authors should provide the segregation of total direct cost on OC as per stage of diagnosis of the disease. This will further strengthen the argument.”

Our response: We totally agree with the reviewer. However, our databases do not allow us to estimate costs by cancer stages. As we explained in the Methods, the Brazilian health information systems are based on ICD-10 codes, which is not categorised by cancer staging. (Methods, Data sources, page 5, line 104-112).

Reviewer 1, comment 11:“The line 249-250 i.e., ‘presentation of the SIA-SUS and SIH-SUS values by number of procedures is an impairment to identifying the individual costs of OC”. Please try to rephrase it for better understanding.”

Our response: Thank you for the comment and the opportunity to clarify this. For better understanding, we rephrased the statement as follows:

‘the SIA-SUS and SIH-SUS data do not allow to identifying the OC costs by individual;’ (Discussion, page 20, lines 331-332)

Reviewer 1, comment 12:“Overall, the discussion section needs improvement in terms of clarity of the language for better understanding of the readers.”

Our response: Thank you for raising this issue. The discussion section was reorganised and the entire article underwent a new linguistic review.

Reviewer 2, comment 1:“The authors presented a paper about direct healthcare costs of lip, oral cavity and oropharyngeal cancer in Brazil. There is a large gap in the scientific literature on studies evaluating the economic impact of cancer treatment, especially in lower-middle- and upper-middle income countries. It is very important to disseminate information about this issues in regions where this information are rare. Particularly for OC, where the late detection enlarge exponentially costs and lives it is very important evaluate its economic impact to built efficient public policies.

I would like to contribute with some questions on intention to clarify some aspects of the paper”

Our response: We would like to thank you for your consideration and time spent reviewing our manuscript.

Reviewer 2, comment 2: “Major issues:Methodology1. Page 6, lines 133-139. Although the authors has explained the source of databases and how they extract the information, it is unclear the choice of this methodology and a theoretical support. When the authors use to calculate TCR the total cost in numerator, they are introducing an important bias in the results because they are treating procedures as individuals, which produce an overestimation of the TCR. We know that the we have linked to an only person a set of procedures to the same treatment.”

Our response: Thank you for your comment. In fact, we used the general population as the denominator, which is also inappropriate. Unfortunately, we have not identified any study or database that estimates the prevalence of OC stratified by anatomical site in Brazil. Based on that, we decided to exclude these estimates from our study. We excluded column 10 from Table 3 and the text related to this calculation in the Methods section. (Results, page 14-15)

Reviewer 2, comment 3: “Major issues:Methodology2.Page 7, lines 140-141. The authors should explain why they choose the Prais-Winsten method to perform linear regression and include more details about the model”

Our response: Thank you for your comment. We have included in the Methods section an explanation as to why the Prais-Winsten method was chosen to perform linear regression as follows: 

‘The trend analysis of OC costs, by anatomical site, was performed using the Prais-Winsten method of linear regression, considering the trend analysis to model the seasonality that is present in the time series data of this study. The Prais-Winsten method is recommended in this type of analysis, especially to deal with the serial autocorrelation present in these cases. The serial autocorrelation can induce misinterpretation resulting from undue significance of minor variations. The smaller the number of points included in the series, the more sensitive this effect will be [20].’ (Methods, Data analysis, page 7, lines 157-162)

Reviewer 2, comment 4: “Major issues:Results1. Page 9, lines 157-160 and Table 1. The parameters of the linear regression on table 1 is insufficient to conclude about the model adjustment and about the significance of the β parameters.”

Our response: Thank you for your comment. The average annual increment rate (AIR) was calculated based on the following equation [Antunes JLF, Cardoso MRA. 2015]:

Annual increment rate= -1+10^b

where b is the slope coefficient obtained in the regression analysis that relates the logarithm of the expenditure with the year of occurrence. The 95% confidence interval of AIR was calculated as [Antunes JLF, Cardoso MRA. 2015]:

IC95%=-1+10^((b±t*EP))

where t is the value at which the Student t distribution showing 16 degrees of freedom at a 95% two-tailed confidence level; EP is the standard error of the estimate of b obtained in regression analysis. The degrees of freedom equal the number of elements in the sample n minus the number of parameters estimated (n-1). In the present study, the number of elements n was equal to the period of 9 years (2008 to 2016) considered in the analysis. Figure 1 presents the results of the Prais-Winsten estimates and the 95% IC for each site of OC considered in our analyses. Different to the others, the C10 – oropharynx coefficient is significant, which shows a tendency in the time series of these costs. It is supposed that this difference is related to the high prevalence of OC at this anatomical site, as commented on by Reviewer 1, comment 6.

Reference:

Antunes JLF, Cardoso MRA. Using time series analysis in epidemiological studies. Epidemiol. Serv. Saúde, Brasília, 2015; 24(3):565-576. 

 Figure 1 - Prais-Winsten estimators of the total costs by anatomical sites

Reviewer 2, comment 5: “Major issues: Discussion1. Page 15, lines 194-202. It is difficult to compare both studies at least to reasons: the thyroid study use a methodology that really estimates a cost by patient. The other reason is despite of both topographies had similar incidence rates, the terapeutic approach is very different which made the costs incomparable.”

Our response: We appreciate the comment and agree that thyroid cancer is not the best comparator to our findings. However, we did not find any specific study on the cost of oral cancer. Therefore, we choose this study to be discussed in parallel with our findings since, besides the similarity in the incidence rate of thyroid cancer, the authors have used the same methodology of aggregate cost from the information systems of the Ministry of Health of Brazil, using ICD-10 codes, a similar period of time, i.e. 2008-2015, and discussed head and neck cancer with a biological behaviour similar to oral squamous cell carcinoma. However, in light of your comment and following a deep discussion within the research group, we decided to reformulate the text as follows: 

‘Differences in the costing method and in the organisation of health systems are two of the main reasons that makes economic evaluation results comparisons a challenge. Evidence has shown an increase in the number of health economic evaluations (HEEs) of cancer in Brazil [4]. However, although oral cancer is one of the five most common neoplasms among males, only 1.8% of the HEEs found were related to the oral cavity or pharyngeal cancer, from 1998 to 2013 [4], and none of them was a cost-of-illness study. We did not find any specific study on the cost of oral cancer. The only cost-of-illness study found in the head and neck region, with epidemiological (incidence rate) and methodological (same methodology of aggregate cost from the information systems of the Ministry of Health of Brazil, using ICD-10 code, 2008-2015) similarities to ours, was related to thyroid cancer [22], for which the therapeutic approach is very different, making the costs incomparable.’ (Discussion, page 16, lines 217-228)

Reviewer 2, comment 6: “Major issues:Discussion2. Page 15, lines 203-206. The authors should take in mind the OC epidemiology. The incidence in males is about 3,5 times higher than in woman and this issue will be reproduced in the healthcare costs”

Our response: Thank you for your comment. We have rephrased the text to better reflect the relationship between the results of the cost of illness in Brazil and the global epidemiological aspects oral cancer, as follows:

‘Even though data collection was not based on individuals, the results reproduce the epidemiological aspects of OC [3,9,10,26, 36-38], since the number of procedures and cost of illness were three times higher for men over 50 years of age than for women’. (Discussion, page 20, lines 313-315)

Reviewer 2, comment 7: “Major issues:Discussion3. Page 17, lines 236-247. The authors should take in mind that regional distribution follow the population distribution and it is not easy to link with the inequalities in public spending between the different regions”

Our response: Thank you for the opportunity to clarify this. The impact of inequalities in public spending between different regions is the subject of important discussions in the Brazilian social economic context. However, we agree that it should not be taken in a generalised context. Thus, this paragraph, which was based on a Brazilian study, was removed from the text.

Reviewer 2, comment 8:Minor issues:Introduction1. Page 3, lines 50-51: Update the information about Brazilian cancer estimates to 2020-2022, like the authors already done in another part of the manuscript.”

Our response: We have updated the estimates in the text as suggested by the reviewer. (Introduction, page 3, lines 50-53)

Proofreading certificate

---

## [Decision Letter · Decision Letter 1]

29 Dec 2020

PONE-D-20-19121R1

Direct healthcare costs of lip, oral cavity and oropharyngeal cancer in Brazil

PLOS ONE

Dear Dr. ROTTA,

Thank you for submitting your manuscript to PLOS ONE. After careful consideration, we feel that it has merit but does not fully meet PLOS ONE’s publication criteria as it currently stands. Therefore, we invite you to submit a revised version of the manuscript that addresses the points raised during the review process.

We look forward to receiving your revised manuscript.

Kind regards,

Gabriel A. Picone

Academic Editor

PLOS ONE

Reviewers' comments:

Reviewer's Responses to Questions

**Comments to the Author**

1. If the authors have adequately addressed your comments raised in a previous round of review and you feel that this manuscript is now acceptable for publication, you may indicate that here to bypass the “Comments to the Author” section, enter your conflict of interest statement in the “Confidential to Editor” section, and submit your "Accept" recommendation.

Reviewer #1: All comments have been addressed

Reviewer #2: All comments have been addressed

2. Is the manuscript technically sound, and do the data support the conclusions?

Reviewer #1: Yes

Reviewer #2: Yes

3. Has the statistical analysis been performed appropriately and rigorously? 

Reviewer #1: Yes

Reviewer #2: Yes

4. Have the authors made all data underlying the findings in their manuscript fully available?

Reviewer #1: Yes

Reviewer #2: Yes

5. Is the manuscript presented in an intelligible fashion and written in standard English?

Reviewer #1: No

Reviewer #2: Yes

6. Review Comments to the Author

Reviewer #1: The authors have done a good work in terms of providing better explanation of the procedures/methodology followed and a background on the health information system in the methods section of the manuscript. But, I feel that the discussion can still be improved in terms of better structuring, content and flow. Below are few suggestions:

1. The issue (mentioned on line 278-281) of presentation of results in USD by other studies cannot be stated as a limitation of the present study. Firstly, it is not a limitation (of the methodology followed) of the present study. Secondly, authors can easily adjust the estimates from US$ to I$ based on the latest conversion and inflation rates. So, I think that the authors should try to report the results of the ‘other small number of studies’ (by adjusting the estimates to I$) on OC in the discussion section.

2. The paragraphs are never three lines short. However, the paragraphs on lines 197-199, 259-161 and 262-266 are of just 3-4 lines. They can either be merged with the other paragraphs maintaining the flow of the discussion or can be deleted strategically without breaking the flow.

3. The paragraph from lines 262-266 does not fit in the flow the discussion section. It could either be deleted or merged in the other paragraph on line 282-292. Similarly, the paragraph with lines 229-236, can be reduced (by summarizing the information presented in it) and merged with the subsequent paragraph.

4. Information mentioned in the last 2 paragraphs (of the discussion section) specifically focuses on strategies to reduce the incidence of cancers. This information is quite important, but authors can try to summarize in a better and intelligible fashion in a short informative paragraph.

Reviewer #2: The authors made satisfactory explanations, accepted the revisor suggestions and improve the paper make them better.

7. PLOS authors have the option to publish the peer review history of their article (what does this mean?). If published, this will include your full peer review and any attached files.

Reviewer #1: No

Reviewer #2: No

---

## [Author Response · Author response to Decision Letter 1]

6 Jan 2021

Dear Editor,

Thank you for considering a revised version of our manuscript PONE-D-20-19121 entitled ‘Direct healthcare costs of lip, oral cavity and oropharyngeal cancer in Brazil’. All addressed points raised during the review process were considered and the manuscript changes are presented below. The manuscript line numbers cited on each answer below are in accordance with the file “manuscript_with_track_changes”.

Yours sincerely, 

The Authors

Reviewer 2, comment 1: “1.The issue (mentioned on line 278-281) of presentation of results in USD by other studies cannot be stated as a limitation of the present study. Firstly, it is not a limitation (of the methodology followed) of the present study. Secondly, authors can easily adjust the estimates from US$ to I$ based on the latest conversion and inflation rates. So, I think that the authors should try to report the results of the ‘other small number of studies’ (by adjusting the estimates to I$) on OC in the discussion section. ”

Our response: Thank you for raising this question and for the opportunity to clarify and fix it. In fact, in the paragraph from lines 278-281 we intended to highlight limitations of our study but also, other aspects to be considered for further studies, which were not a limitation of our study. However, the text was not clear not only about that, but also regard the “small number of studies”, which were already discussed on the first paragraph of the discussion and their results were reported on the second paragraph of discussion section (lines 213-224) adjusting their estimates to I$. The paragraph from lines 278-281 was rephrased (lines 296-299).

Reviewer 2, comment 2: The paragraphs are never three lines short. However, the paragraphs on lines 197-199, 259-161 and 262-266 are of just 3-4 lines. They can either be merged with the other paragraphs maintaining the flow of the discussion or can be deleted strategically without breaking the flow. 

Our response: We agree with the suggestions and changes in the text were done aiming to maintain the flow of the discussion, which can be seen on lines: 199-212; 232-240; 296-299.

Reviewer 2, comment 3: The paragraph from lines 262-266 does not fit in the flow the discussion section. It could either be deleted or merged in the other paragraph on line 282-292. Similarly, the paragraph with lines 229-236, can be reduced (by summarizing the information presented in it) and merged with the subsequent paragraph.

Our response: Changes on paragraph from lines 262-266 can be seen on lines 271-276.The paragraph from lines 229-236 was merged with subsequent paragraph and summarized into the paragraph from lines 232-240.

Reviewer 2, comment 4: Information mentioned in the last 2 paragraphs (of the discussion section) specifically focuses on strategies to reduce the incidence of cancers. This information is quite important, but authors can try to summarize in a better and intelligible fashion in a short informative paragraph. 

Our response: The last two paragraphs of discussion were summarized in a shorter paragraph from lines 300-309 and one reference (43*) was added to strength scientific support.

Reference 43*: Brocklehurst P, Kujan O, O'Malley LA, Ogden G, Shepherd S, Glenny AM. Screening programmes for the early detection and prevention of oral cancer. Cochrane Database Syst Rev. 2013;19(11):CD004150. doi: 10.1002/14651858.CD004150.

---

## [Decision Letter · Decision Letter 2]

20 Jan 2021

Direct healthcare costs of lip, oral cavity and oropharyngeal cancer in Brazil

PONE-D-20-19121R2

Dear Dr. ROTTA,

We’re pleased to inform you that your manuscript has been judged scientifically suitable for publication and will be formally accepted for publication once it meets all outstanding technical requirements.

Kind regards,

Gabriel A. Picone

Academic Editor

PLOS ONE

Additional Editor Comments (optional):

Reviewers' comments:

Reviewer's Responses to Questions

**Comments to the Author**

1. If the authors have adequately addressed your comments raised in a previous round of review and you feel that this manuscript is now acceptable for publication, you may indicate that here to bypass the “Comments to the Author” section, enter your conflict of interest statement in the “Confidential to Editor” section, and submit your "Accept" recommendation.

Reviewer #1: All comments have been addressed

2. Is the manuscript technically sound, and do the data support the conclusions?

Reviewer #1: Yes

3. Has the statistical analysis been performed appropriately and rigorously? 

Reviewer #1: Yes

4. Have the authors made all data underlying the findings in their manuscript fully available?

Reviewer #1: Yes

5. Is the manuscript presented in an intelligible fashion and written in standard English?

Reviewer #1: Yes

6. Review Comments to the Author

Reviewer #1: The authors have adequately addressed all the comments raised in the review. The authors have done a good work in terms of providing better explanation of the procedures/methodology followed and improving the discussion section of the manuscript.

7. PLOS authors have the option to publish the peer review history of their article (what does this mean?). If published, this will include your full peer review and any attached files.

Reviewer #1: No

---

## [Editor Report · Acceptance letter]

2 Feb 2021

PONE-D-20-19121R2 

Direct healthcare costs of lip, oral cavity and oropharyngeal cancer in Brazil 

Dear Dr. ROTTA:

I'm pleased to inform you that your manuscript has been deemed suitable for publication in PLOS ONE. Congratulations! Your manuscript is now with our production department. 

Kind regards, 

on behalf of

Dr. Gabriel A. Picone 

Academic Editor

PLOS ONE